# Structural Basis for the Interaction between the Ezrin FERM-Domain and Human Aquaporins

**DOI:** 10.3390/ijms25147672

**Published:** 2024-07-12

**Authors:** Helin Strandberg, Carl Johan Hagströmer, Balder Werin, Markus Wendler, Urban Johanson, Susanna Törnroth-Horsefield

**Affiliations:** Department of Biochemistry and Structural Biology, Lund University, 221 00 Lund, Sweden; helin.strandberg@biochemistry.lu.se (H.S.); carl.johan.hagstromer@ki.se (C.J.H.); balder.werin@biochemistry.lu.se (B.W.); markus.wendler@med.lu.se (M.W.); urban.johanson@biochemistry.lu.se (U.J.)

**Keywords:** Aquaporin 2, Aquaporin 5, ezrin, FERM, protein–protein interactions, microscale thermophoresis, in silico modelling, membrane protein trafficking

## Abstract

The Ezrin/Radixin/Moesin (ERM) family of proteins act as cross-linkers between the plasma membrane and the actin cytoskeleton. This mechanism plays an essential role in processes related to membrane remodeling and organization, such as cell polarization, morphogenesis and adhesion, as well as in membrane protein trafficking and signaling pathways. For several human aquaporin (AQP) isoforms, an interaction between the ezrin band *F*our-point-one, *E*zrin, *R*adixin, *M*oesin (FERM)-domain and the AQP C-terminus has been demonstrated, and this is believed to be important for AQP localization in the plasma membrane. Here, we investigate the structural basis for the interaction between ezrin and two human AQPs: AQP2 and AQP5. Using microscale thermophoresis, we show that full-length AQP2 and AQP5 as well as peptides corresponding to their C-termini interact with the ezrin FERM-domain with affinities in the low micromolar range. Modelling of the AQP2 and AQP5 FERM complexes using ColabFold reveals a common mode of binding in which the proximal and distal parts of the AQP C-termini bind simultaneously to distinct binding sites of FERM. While the interaction at each site closely resembles other FERM-complexes, the concurrent interaction with both sites has only been observed in the complex between moesin and its C-terminus which causes auto-inhibition. The proposed interaction between AQP2/AQP5 and FERM thus represents a novel binding mode for extrinsic ERM-interacting partners.

## 1. Introduction

Ezrin is a member of the ERM (Ezrin/Radixin/Moesin) family of proteins that act as dynamic cross-linkers between the actin cytoskeleton network and the plasma membrane. This mechanism is important for organizing and maintaining specialized membrane domains and is achieved by ERM-proteins simultaneously binding to filamentous actin and cytoplasmic domains of transmembrane proteins, membrane-associated proteins or membrane lipids. As such, ERM-proteins are crucial for cell morphogenesis, polarization, motility and adhesion [1,2]. In addition, ERM-proteins play essential roles in membrane protein trafficking as well as the formation of the immune synapse and have been shown to act as signal transducers in membrane-associated signaling pathways [3]. ERM-proteins are highly conserved with human ezrin, radixin and moesin sharing 73–81% sequence identity. The functional diversity between the three proteins is not clear and it has been suggested that they can functionally replace each other [4]. Nevertheless, the fact that they are expressed in a tissue-specific manner and can be differentially phosphorylated suggests that they may carry out related but distinct functions in different cell types [1].

ERM-proteins are composed of three distinct domains: an N-terminal FERM- (band *F*our-point-one, *E*zrin, *R*adixin, *M*oesin) domain that binds numerous transmembrane and membrane-associated proteins, an α-helical coiled-coil linker region, and a C-terminal domain that contains the actin-binding site (Figure 1) [1]. ERM-proteins are known to exist in at least two physiologically relevant states: a dormant state where the binding sites are masked by the C-terminal domain interacting with FERM, and an active state in which the sites are accessible [5]. Activation has been proposed to occur in two steps, with FERM first binding the membrane lipid phosphatidyl-inositol 4,5-bisphosphate (PIP_2_) followed by phosphorylation of a conserved threonine in the C-terminal domain [6]. This triggers a conformational change whereby the C-terminus is released, exposing the membrane protein and actin binding sites on both FERM and the C-terminal domain [5,7].

The FERM-domain consists of ~300 amino acids and has the highest sequence identity among the three human ERM-proteins (86%). The crystal structures of this domain from all three ERM-proteins show a highly conserved structure with three sub-domains (F1–F3) that are arranged as a compact clover-shaped molecule (Figure 2a) [8,9,10]. FERM has been shown to directly interact with the cytoplasmic tails of a number of transmembrane proteins, including the adhesion molecules CD43 [11], CD44 [11,12] and ICAM 1-3 [11,13,14], the α_1B_-adrenergic receptor [15] and Na^+^/K^+^-ATPase [16]. Crystal structures of FERM in complex with peptides from several interacting proteins have revealed the presence of a binding site in a groove between α1 and β5 at the edge of sub-domain F3, with the peptide contributing a beta strand to the β5–β7 sheet in an anti-parallel fashion (Figure 2b) [17,18,19,20,21]. In addition, FERM also interacts with multiple transmembrane proteins indirectly via adaptor proteins such as the Na^+^/H^+^ regulatory factor (NHERF) 1 and 2 (also known as ERM-binding phosphoprotein 50/EBP50 and NHE3 kinase A regulatory protein/E3KARP) [22,23]. This involves a distinct binding site at the opposite side of F3, with an amphipathic α-helix from the interacting protein binding in a hydrophobic groove between the β1–β4 and β5–β7 beta sheets (Figure 2b) [24,25]. Both these binding sites are masked in the dormant ERM-protein, as shown by the structures of complexes between FERM and the C-terminal domain [5,7] as well as the structure of full-length dormant moesin from *Spodoptera frugiperda* [26] (Figure 2c).

Aquaporins (AQP) are a highly conserved family of membrane-integral water channels that facilitate the flux of water, as well as other small solutes such as glycerol and urea, across cellular membranes. In mammals, thirteen isoforms have been identified that are expressed in a tissue-dependent manner and are involved in many fundamental physiological processes, including urine concentration, exocrine fluid secretion, and skin hydration [27]. The majority of human AQPs have been shown to be regulated by controlling the number of AQP molecules in the plasma membrane through trafficking. The best characterized example of this is the vasopressin-dependent trafficking of AQP2 in the kidney collecting duct which controls the water permeability of the apical membrane and thereby urine volume. This is governed by multiple post-translational modification sites in the AQP2 C-terminus that directs the protein between cellular compartments by altering the affinity towards proteins within the vesicular trafficking machinery. Although not as well characterized, post-translational modification of the cytoplasmic tails of other AQPs also controls their sub-cellular localization in an isoform-dependent manner, allowing for cell- and tissue-specific AQP regulation [28].

Several mammalian AQP isoforms have been proposed to directly interact with Ezrin during trafficking [29,30,31]. For AQP0, the major intrinsic membrane protein of the eye lens, cross-linking mass spectrometry on the membrane fraction from bovine lens cortex as well as an in vitro Glutathione S-transferase (GST)-pulldown assay demonstrated an interaction between the Ezrin FERM-domain and the AQP0 C-terminal tail [29]. It was suggested that this interaction may be responsible for anchoring the EPPD-complex (*E*zrin, *P*eriplakin, *P*eriaxin, *D*esmoyokin), a cell–cell junction system found in lens fiber cells [32], to the plasma membrane. A direct interaction between the ezrin FERM-domain and the C-terminal tail was also shown for AQP2 by co-immunoprecipitation (co-IP) and pull-down assays and Ezrin knockdown in a porcine kidney cell line (LLC-PK1) resulted in significant accumulation of AQP2 in the apical membrane and reduced AQP2 endocytosis [30]. Moreover, moesin was implicated in apical targeting of AQP2 in CD8-cells, although the mechanism was not defined [33]. Finally, co-IP and a proximity ligation assay (PLA) in a salivary gland cell line (NS-SV-AC) identified an interaction between ezrin and AQP5 and it was hypothesized that this interaction also involves the FERM-domain, although this was not verified experimentally [31]. In mouse lung epithelial cells (MLE-12), inhibition of ezrin selectively suppressed AQP5 translocation to the membrane in response to a Ca^2+^-ionophore [34]. Furthermore, in salivary gland biopsies from patients with Sjögren’s syndrome, an autoimmune disease in which the destruction of exocrine glands leads to severe eye and mouth dryness, AQP5 and ezrin were both mislocalized, supporting the importance of ezrin for AQP5 trafficking [31].

In this study, we aimed to explore the molecular basis for the interaction between ezrin and human AQPs and elucidate if there is a common structural mechanism for how these proteins interact. Using MST, we confirm that human AQP2 and AQP5 directly interact with the Ezrin FERM-domain via their C-terminal tails, as previously proposed [30,31]. For both AQPs, full-length protein and the C-terminal tails interacted with FERM with dissociation constants (K_d_) in the low micromolar range. In silico modelling of the complexes suggests a common mode of binding whereby the respective AQP C-terminus interacts with two regions of the F3-subdomain, thereby occupying both sites that had previously been shown to individually bind other FERM-interacting proteins. The simultaneous binding of both F3-sites had previously only been seen in the dormant FERM-C-terminus complex and represents a novel mode of binding for ERM-interacting proteins.

## 2. Results

### 2.1. Full-Length AQP2 and AQP5 Bind FERM with Similar Affinity

While studies in cells as well as pull-down assays have suggested that Ezrin binds to several AQPs via its FERM-domain [29,30,31], the direct interaction between the proteins remains to be conclusively established. To verify this as well as to obtain quantitative information regarding binding affinities, we studied the interaction between FERM and recombinantly produced and purified full-length AQP2 and AQP5 using MST. MST detects complex formation from a change of movement in a thermal gradient and has previously been used in our laboratory to study a number of AQP-complexes [35,36,37,38].

For each AQP, a 1:1 dilution series was made resulting in 12 samples with a concentration range of 0.0419–86.0 μM for AQP2 and 0.029–60.6 μM for AQP5. The samples were mixed with an equal volume of FERM that had been labelled with Alexa-488, transferred to MST capillaries, and MST traces were recorded (Appendix A). From these traces, the difference in normalized fluorescence before and after heating was used to generate a binding curve for each AQP from which the dissociation constant (K_d_) could be determined (Figure 3a). Both AQPs bound FERM with similar affinity: K_d_ was 7.8 ± 3.8 and 14 ± 5.7 μM for AQP2 and AQP5, respectively.

In the capillaries with the highest AQP concentrations the initial fluorescence deviated significantly from the mean, wherefore these could not be included in the MST-analysis. This artefact became even more prominent when the AQP concentration was increased further. As a result, AQP concentrations that would lead to saturation could not be used in the case of AQP5, wherefore the estimated K_d_-value should be evaluated with caution.

### 2.2. The Interaction with FERM Is Mediated by the AQP C-Terminal Tails

Next, we set out to investigate if the interaction between FERM and AQP2/AQP5 is mediated by the AQP C-termini as previously proposed [30,31]. For this purpose, we expressed the AQP2 and AQP5 C-terminal tails (residues 227–271 and 228–265 for AQP2 and AQP5 respectively) as GST-fusion constructs in *E. coli*. The purified GST-AQP constructs were then cleaved with thrombin, allowing AQP2 and AQP5 C-termini to be separated from the GST-tag. For each AQP C-terminal peptide, a 1:1 dilution series was made resulting in 12 samples for AQP2 (0.072–294.5 μM) and 16 samples (0.001–337.5 μM) for AQP5, mixed with fluorescently labelled FERM, and MST-traces were recorded and analyzed as described above (Appendix A). As seen in Figure 3b, both the AQP2 and AQP5 C-termini bound FERM. For the AQP2 C-terminal peptide, K_d_ was estimated as 8.7 ± 2.4 µM whereas the affinity for the AQP5 C-terminal peptide was somewhat higher (*p* < 0.05), K_d_ was 2.9 ± 0.89 µM. This confirms the C-terminus as the region responsible for the FERM-interaction for both AQP2 and AQP5.

### 2.3. In Silico Modelling Supports a Common Mode of Binding

The amino acid sequences of the AQP2 and AQP5 C-termini are highly similar and contain a structurally conserved amphipathic helix that is known to be involved in interactions with regulatory proteins (Figure 4a,b) [39,40]. We therefore hypothesized that there may be a common binding mode for how these proteins interact with FERM. To explore this, we modelled the AQP2- and AQP5-FERM complexes using ColabFold (version 1.3.0) [41]. First, the complexes between full-length AQPs and FERM were generated (Figure 4c and Appendix A). In these models, the distal C-terminus (residues 254–258 and 257–261 for AQP2 and AQP5 respectively) formed a β-strand that bound to the β1–β5 sheet in the FERM subdomain F3 in an antiparallel manner. Specifically, Val 257 and Leu 259 from AQP2 and the corresponding residues Met 260 and Leu 262 from AQP5 fit into a hydrophobic groove formed between α1 and β5 in F3, while hydrophilic residues on the opposite side of the peptide β-strand are exposed to the solvent (Figure 4d). This is highly reminiscent of how adhesion proteins such as CD44 and ICAM-2 bind FERM (Figure 2b) [17,18], thus supporting the mode of interaction observed in the predicted models.

To our surprise, the proximal C-terminus which forms the conserved amphipathic helix that is believed to be a common protein–protein interaction site did not participate in the interaction. When we took a closer look at how this region was modelled by ColabFold, we noticed that this helix was consistently placed across the cytoplasmic interface of the AQP monomer, as commonly observed in AQP crystal structures. However, from the crystal structure of human AQP2, it is known that this region is highly flexible with the amphipathic α-helix adopting four different positions in the tetramer (Figure 4b) [42], none of which is represented by the ColabFold model. Moreover, in the case of the binding of calmodulin (CaM) to AQP0, the helix has been proposed to detach from this location so that the helices from two different monomers bind one CaM-molecule simultaneously [39]. To create a scenario which avoids the proximal C-termini being fixed in a position where it is not available for the interaction, we therefore also generated models of FERM in complex with only the AQP2 or AQP5 C-termini (Figure 5 and Appendix A). In both these models, the proximal and distal parts of the AQP C-termini simultaneously bind to the FERM F3 subdomain at distinct sites (Figure 5a). While the distal C-termini interacts in the same manner as in the full-length AQP complexes (Figure 4), an additional interaction with FERM is observed involving the amphipathic α-helix formed by the proximal part of the C-terminus (residues 230–237 and 232–239 for AQP2 and AQP5 respectively). Similarly, as for the distal C-termini, the binding mode proposed by ColabFold closely resembles what has been observed in crystal structures of FERM-complexes also for this second site [24,25]. In both AQP-FERM models, hydrophobic residues on one side of the amphipathic α-helix bind in the same groove on F3 (Figure 5b) and in a similar manner as seen in the FERM-NHERF-1 and 2 crystal structures (Figure 2b). It thus seems that the interaction between FERM and these two AQPs may involve a common mechanism whereby the AQP C-termini occupy both previously identified binding sites on the FERM F3 subdomain. The simultaneous binding to both these sites has previously only been seen in the dormant state of ERM-proteins in which the interaction between FERM and the ERM C-terminal domain causes auto-inhibition (Figure 2c) [5,7,26]. The models presented here thus represent a novel mode of binding for extrinsic FERM-interacting partners.

## 3. Discussion

In eukaryotic cells, actin filaments together with other cytoskeletal proteins form a layer beneath the plasma membrane that is known as the cell cortex. The cell cortex provides structural support for the plasma membrane but is also vital for many dynamic membrane-related processes, including those involved in vesicle and membrane protein trafficking. ERM-proteins coordinate these processes by providing a regulated linkage between the plasma membrane and the actin network and have been shown to play important roles in endo- and exocytosis as well as recycling of several membrane proteins. For the α1b-adrenergic receptor, the direct interaction with Ezrin contributes to receptor recycling and exocytosis [15]. This involved a poly-arginine motif on the α1b C-terminus which was proposed to bind the FERM-domain in a similar manner as the adhesion protein ICAM-2 (Figure 2b). Moreover, the interaction between Syntaxin-3 and Ezrin is essential for the membrane insertion of H^+^-K^+^-ATPase in gastric parietal cells and it has been suggested that this is controlled by phosphorylation-dependent conformational changes within Ezrin [43,44]. Ezrin also indirectly associates with receptors and channels via PDZ-domains on adaptor proteins such as NHERF-1 and 2.

For the cystic fibrosis transmembrane conductance regulator (CFTR) and the Na^+^-H^+^ exchanger 3 (NHE3), the NHERF-mediated binding to ERM-proteins has been shown to facilitate exocytosis and plasma membrane delivery [45,46]. Interestingly, NHERF also promotes the assembly of CFTR-NHE3 complexes, the stoichiometry of which is modulated by Ezrin [47]. The ability of NHERF to bind multiple other membrane proteins and the existence of additional adaptor proteins with PDZ-domains further expands the repertoire of proteins that is likely to associate with ERM-proteins. These studies strongly support a role for ERM-proteins in clustering membrane proteins and membrane-associated proteins during vesicular trafficking and the assembly of signaling complexes [1].

In the case of AQPs, there are several studies that suggest that a direct interaction with ezrin is involved in trafficking to and from the plasma membrane (see Section 1) [29,30,31,33,34]. The results presented here add further support to this and give new insights into the molecular details of this process. Using recombinantly expressed proteins, we show for the first time the direct interaction between FERM and AQP2 as well as AQP5. Furthermore, as previously proposed for AQP0 and AQP2 from pull-down assays [29,30], we show that AQP2 and AQP5 directly bind FERM via their C-terminal tails (Figure 3), thus confirming our previous hypothesis [31]. Our studies provide the first quantitative analysis of the AQP-FERM interaction and reveal affinities in the low micromolar range for all constructs (K_d_ is 7.8 ± 3.8/14 ± 5.7 μM for full-length AQP2/AQP5 and 8.7 ± 2.4/2.9 ± 0.89 μM for the AQP2/AQP5 C-terminal tails). While the affinity for full-length AQP2 and the AQP2 C-terminal tail is highly similar, there is a significant difference between full-length AQP5 and the AQP5 C-terminal tail (*p* < 0.05). One possible explanation for this could be a lower conformational flexibility of the AQP5 C-terminus compared to AQP2, as suggested from their crystal structures [42,48]. In AQP2, a double proline motif immediately prior to the amphipathic helix that is proposed to bind to one of the binding sites on FERM (Figure 6) is believed to act as a hinge region, allowing for a significant positional variability that is not observed for AQP5 where only one proline is found at the equivalent position (Figure 4a,b).

Previous binding studies on FERM and interacting peptides have given a wide variety of affinities, with K_d_-values ranging from low nM to a few μM [15,17,18,21,24]. The affinity observed for the AQP2 and AQP5 C-termini are most similar to those obtained for ICAM-3 (0.75 μM) [17], the α1b C-terminus (~0.2 μM) [15] and the radixin-neutral endopeptidase 24.11 (NEP) complex (2.2 μM) [21], all of which are proposed to bind as a complementary β-strand to the β1–β5 sheet in F3 (Figure 2a). For NEP in particular, the sequence of the interacting β-strand is highly similar to the corresponding region in the AQP models, and this has been suggested as an alternative binding motif compared to the consensus motif proposed for adhesion molecules such as ICAM 1-3 [21] (Figure 6).

As for the second binding site on FERM, peptides that interact contain an amphipathic α-helix that binds with its hydrophobic side in a cleft on the FERM F3 subdomain, as seen in the crystal structures of the complexes between FERM and NHERF-1/-2 [24] (Figure 2b). Based on this, a signature sequence for this interaction was proposed (Figure 6a). While the AQP C-termini do not share all elements of this signature motif, the region that occupies this binding site in the FERM-AQP C-terminal peptide models does form an amphipathic helix that has strong similarities to those found in NHERF (Figure 6b). Furthermore, in the complex between human moesin and its C-terminus (Figure 2b), the sequence of the α-helix bound at this site is highly similar to the corresponding sequence in the AQP C-termini (Figure 6a) [5]. The same amphipathic α-helix has been shown to be involved in the interaction between AQPs and several other proteins that are involved in their regulation, including AQP0 and CaM [39], AQP2 and LIP5 [40] and AQP4 and CaM [37]. For AQP2 and LIP5, a docking model supported by mutagenesis studies revealed a binding mode that is highly reminiscent of the interaction proposed here (Figure 6b) [40].

While it could be argued that the AQP C-termini do not exist in isolation, there are several reasons why we believe that the interaction seen in FERM-AQP C-terminal peptide models is more likely to represent the true binding mode. First, we know from the crystal structure of AQP2 that the linker region between the last transmembrane helix and the amphipathic helix is flexible and that ColabFold models this part of the C-terminus in a position that does not agree with any of the experimentally determined positions (Figure 4b) [42]. Secondly, there is compelling evidence that the interactions between AQP2 and LIP5 as well as between AQP0 and CaM involves the hydrophobic side of the amphipathic helix in the proximal C-terminus [39,40], interactions which would not be possible without the helix detaching from its position in the full-length models. Finally, analysis of cross-linked peptides from the membrane fraction of lens cortex identified several peptides that support the binding of the AQP0 C-terminal tail to the binding site on FERM which is occupied by the amphipathic helix in the C-terminal peptide models [29]. Taken together, this strongly supports the involvement of the amphipathic helix as proposed by the models of FERM in complex with the AQP C-termini (Figure 5 and Figure 6), suggesting that this is not simply an artefact of not using the full-length protein during the modelling.

In a previous study, we used TrRosetta to predict the structure of the AQP5 C-terminus and perform in silico docking in PyRosetta and HADDOCK 2.4 [31]. In the resulting model, the C-terminus formed a broken α-helix that bound to the same site as NHERF, but in the opposite direction (Figure 6b). While there are similarities between this mode of binding and the signature motif proposed from the FERM-NHERF crystal structures, we believe that the models presented here are more likely to represent the correct AQP-FERM structures. The main reasons for this are: (1) we obtain the same mode of binding for both AQPs, despite minor differences in sequence, (2) the observed interactions closely resemble those observed for other FERM-interacting proteins, and (3) the amphipathic helix, which is a known common AQP protein–protein interaction site, binds with its hydrophobic side to FERM, as proposed for other AQP regulatory complexes. In the previous docking model, the amphipathic helix is not conserved and its hydrophobic residues are partly exposed to the environment. Instead, the more distal C-terminus forms an α-helix that interacts with FERM via hydrophilic residues (Figure 6b).

Although the models presented here display striking similarities with other FERM complexes, there is one significant difference: our models of the AQP C-termini in complex with FERM suggest simultaneous binding to both binding sites on FERM (Figure 5). This has previously been observed for the interaction between FERM and the ERM C-terminal domain [5,26] but has never been suggested for an extrinsic binding partner. It should be noted that all structural studies on FERM-complexes with extrinsic binding partners as well as most binding studies have been conducted with short peptides [15,17,18,19,20,21,24,49]. It is therefore possible that interactions with both binding sites also occur in other complexes but that these have been missed due to construct design. Nevertheless, SPR studies on radixin FERM showed that ICAM-2 and NHERF-peptides, which interact with the two different sites, were not able to bind simultaneously [24]. Based on this, it was proposed that that there is a direct competition between adhesion proteins such as ICAM-2 and adaptor proteins such as NHERF that bridge the interaction with multiple membrane proteins and that this is mediated through conformational changes within the F3 subdomain.

The involvement of both the proximal and distal parts of the AQP C-terminus in regulatory protein–protein interactions has previously been suggested from binding studies of AQP2 and LIP5 [36]. In this study, full-length AQP2 had higher affinity for LIP5 than a construct that had been truncated immediately after the amphipathic helix, whereas peptides corresponding to the only the distal C-terminus did not bind at all. Based on this, it was suggested that the interaction with the amphipathic helix is necessary for the ability of the distal C-terminus to interact. Moreover, introduction of phospho-mimicking mutations at the sites within the distal C-terminus that are responsible for controlling vasopressin-dependent trafficking of AQP2 modulated the affinity in a site-specific manner. In this respect it is interesting to note that Ser256, the major phosphorylation site for governing targeting of AQP2 to the apical membrane, is located on the β-sheet that interacts with FERM (Figure 4d). Further studies will be needed to confirm if the interaction between AQPs and FERM are indeed phosphorylation-dependent. 

## 4. Materials and Methods

### 4.1. Cloning, Expression and Purification of the FERM-Domain from Human Ezrin

The *E. coli* codon-optimized gene for the human ezrin FERM-domain (residues 1–295) was synthesized (Genscript, Piscataway, NJ, USA) and cloned into pET15b using the NdeI and BamHI sites. The final construct contained an N-terminal 8xHis-tag followed by a TEV cleavage site. The plasmid was transformed into calcium-competent *E. coli* cells BL21* by heat shock. 2x1L LB-media supplemented with 100 μg/mL ampicillin was inoculated with a 5 mL overnight culture. The cultures were placed in a shaking incubator at 37 °C for 2.5 h until OD600 was 0.4–0.6 after which the protein expression was induced by adding IPTG to a final concentration of 0.5 mM. The expression was performed at 30 °C in a shaking incubator at 100 rpm overnight. The following day, cells were harvested by centrifugation at 8000 rpm for 30 min and the pellets were combined and stored at −20 °C.

The thawed cells were resuspended in 50 mL Lysis buffer (20 mM Tris (pH 7.4), 250 mM NaCl, 5% glycerol, 1 mM DTT) supplemented with one cOmplete™ EDTA free protease cocktail tablet (Roche, Basel, Switzerland). The cells were broken by sonication (10 × 1 min with 1 min pauses in between) while kept on ice. After spinning down the cell debris at 10,000× *g* for 30 min, the supernatant was supplemented with 10 mM imidazole and loaded on a His-trap column (Cytiva Life Sciences, Marlborough, MA, USA). The His-trap column was equilibrated with lysis buffer before loading the sample and was then washed with buffer supplemented with 75 mM imidazole before eluting the protein in buffer containing 300 mM imidazole. The eluted protein fractions were analyzed on an SDS-PAGE gel stained with Coomassie brilliant blue and suitable fractions were pooled together and stored at −20 °C.

### 4.2. Expression and Purification of Full-Length AQPs

The full-length AQP constructs consist of AQP2 with an uncleavable His-tag attached to the N-terminus and untagged AQP5, both codon-optimized for production in Pichia pastoris. For both AQPs, expression and purification were performed as previously described [42,48]. Briefly, cells were cultivated in basal salt media within a 3 L fermenter (Belach Bioteknik AB, Skogås, Sweden). Upon glycerol depletion, an additional 200 mL of glycerol was supplied to promote cell growth. Subsequently, cells underwent gradual methanol feeding for 24 h to induce protein expression. Cell harvesting was performed at 6000× *g* for 20 min at 4 °C, yielding 250–500 g cells/L of culture.

For the preparation of the membrane, 50–100 g of cells was thawed and suspended in a phosphate buffer containing 50 mM potassium phosphate at pH 7.5, 2 mM EDTA, and 5% glycerol. Lysis was then performed, with 1 mM PMSF supplemented before utilizing a bead beater. The resulting lysate was centrifuged to eliminate cell debris and the supernatant underwent ultracentrifugation to isolate crude membranes. Membranes were subsequently homogenized and washed with urea buffer composed of 4 M urea, 5 mM Tris pH 9.5, and 2 mM EDTA. After a second wash step with membrane buffer (20 mM Tris pH 8, 20 mM NaCl, 10% glycerol, 2 mM EDTA, and 1 mM PMSF), the membranes were resuspended in the same buffer without EDTA and PMSF at a concentration of 0.5 g membrane/mL, flash-frozen in liquid nitrogen, and stored at −80 °C for future use.

Thawed membranes of AQP2 were solubilized in a 1:1 ratio with solubilization buffer (20 mM Tris pH 8, 300 mM NaCl, 10% glycerol, 4% n-Nonyl-Beta-D-Glucopyranoside (NG, Anatrace, Maumee, OH, USA)). Similarly, AQP5 membranes were solubilized in a solution containing 40 mM MES at pH 6.0, 6% NG, 50 mM NaCl, 10% glycerol, and 2 mM βMeOH. One cOmplete™ EDTA-free protease inhibitor cocktail tablet (Roche, Basel, Switzerland) was added to each mixture. The mixtures were left to solubilize for one hour at 4 °C on a tipping table. Non-solubilized material was removed by centrifugation.

For AQP2, the supernatant was supplemented with 10 mM imidazole before loading on a HisTrap column equilibrated with a buffer containing 20 mM Tris pH 8, 300 mM NaCl, and 0.2% NG. The column was then washed with a buffer supplemented with 75 mM imidazole prior to elution with 300 mM imidazole.

For AQP5, the supernatant was loaded onto and purified utilizing a Resource S column (GE Healthcare, Chicago, IL, USA) in a buffer containing 20 mM MES pH 6.0, 0.4% NG, and 15 mM–1 M NaCl.

Fractions associated with the elution peaks were respectively pooled for both AQPs and concentrated using 30 kDa cut-off Vivaspin concentration tubes (GE Healthcare, USA). The concentrated samples were then loaded on an SEC column (Superdex 200 10/300, GE Healthcare, USA), equilibrated with a buffer containing 20 mM Tris pH 8, 300 mM NaCl, 10% glycerol, and 0.2% NG for AQP2 and equilibrated with a buffer containing 20 mM Tris pH 7.4, 100 mM NaCl, and 0.4% NG for AQP5. Fractions containing the AQPs were individually pooled and concentrated as above, followed by flash-freezing with liquid nitrogen and stored at −80 °C for further use.

### 4.3. Cloning, Expression and Purification of GST-AQP C-Termini

The GST-AQP2 C-terminus construct (GST-AQP2) had been cloned previously [35]. For cloning of GST-AQP5 C-terminus, an *E. coli* codon-optimized gene for residues 228–265 was synthesized (Genscript) and cloned into pET41a using the SpeI and AvrII sites, resulting in a GST-AQP5 C-terminus fusion construct (GST-AQP5). The construct contains a thrombin cleavage site (LVPRGS) between GST and the AQP5 C-terminus. The resulting plasmids (pET41a-AQP5 Cterm) were transformed into calcium-competent *E. coli* BL21* by heat-shock. Both GST-AQP constructs were expressed in 500 mL LB medium with 30 µg/mL kanamycin with 3 h induction with 0.5 mM IPTG. The cells were pelleted and resuspended in 5 mL Lysis buffer (20 mM Tris pH 7.4, 250 mM NaCl, 5% glycerol, 1 mM DTT) per g cells. The cells were sonicated and the lysate was purified with a 1 mL GSTrap column (Cytiva Life Sciences, Marlborough, MA, USA) and eluted with GST elution buffer (50 mM Tris pH 8, 20 mM glutathione, 1 mM DTT).

### 4.4. Thrombin Cleavage and Purification of the Isolated AQP C-Termini

The buffer of the GST-AQP2 and GST-AQP5 samples were changed to Thrombin cleavage buffer (20 mM Tris pH 8, 150 mM NaCl, 2.5 mM CaCl_2_) using a PD10 column (Cytiva Life Sciences, Marlborough, MA, USA). Thrombin was added at a final concentration of 1 U/mg protein and the sample was incubated overnight at 4 °C. The AQP C-termini were separated from the cleaved-off GST tag and thrombin by running the sample through a Vivaspin 6 concentrator with 10,000 kDa MWCO (Cytiva Life Sciences, Marlborough, MA, USA) and collecting the flowthrough. The buffer was changed to Binding assay buffer (20 mM Tris pH 8, 100 mM NaCl, 5% Glycerol, 1 mM DTT) on a PD10 column and the samples were concentrated using a Vivaspin 6 concentrator with 5000 kDa MWCO (Cytiva Life Sciences, Marlborough, MA, USA). For AQP2, the sample was run in a SpeedVac (Thermo Fischer Scientific, Waltham, MA, USA) and the resulting pellet was diluted with water to 1.52 mg/mL or 294.5 µM. As a consequence, the concentration of the buffer components in the final AQP2 sample was twice that of the binding assay buffer.

### 4.5. Fluorescent Labelling of FERM

The FERM-domain was labelled with cysteine-reactive C5 maleimide-Alexa 488 according to the protocol from the manufacturer (Invitrogen probe manual, Waltham, MA, USA). To prevent cysteine bridges, 50 mM TCEP was added to FERM and the sample was incubated for 15 min at room temperature. Alexa-488 and FERM were mixed at a 10:1 molar ratio and incubated on a tipping table at 4 °C overnight, while being protected from light using aluminum foil. The following day, the sample was added to a PD10 desalting column equilibrated with 20 mM Tris (pH8), 100 mM NaCl and 0.5 mM DTT to separate labelled FERM from free Alexa488. The eluted FERM-Alexa488 was concentrated using a spin filter protein concentrator with a 10 kDa cutoff (Thermo Scientific, Waltham, MA, USA), aliquoted and stored at −20 °C.

### 4.6. Microscale Thermophoresis

A 1:1 dilution series was prepared in binding buffer for both full-length AQPs and AQP C-terminal peptides, resulting in 12 samples each for the full-length constructs and the AQP2 C-terminal peptide and 16 samples for the AQP5 C-terminal peptides, respectively. Following mixing with an equal volume of FERM-Alexa488, the samples covered a concentration range of 0.0419–86.0 μM and 0.029–60.6 μM for full-length AQP2 and AQP5, respectively. For the C-terminal peptides, concentrations ranged from 0.072–294.5 μM for AQP2 and 0.001–337.5 μM for AQP5. The final concentration of FERM-Alexa488 was 0.125 μM for all constructs except for the AQP5 C-terminus, where 0.129 μM was used. Subsequently, the samples were transferred to premium (full-length AQP) or standard (C-terminal peptides) MST capillaries and MST-traces were recorded on a Monolith NT.115 (NanoTemper Technologies GMBH, Munich, Germany) at room temperature, with LED and MST-powers set to 80%. MST data were recorded for three individually prepared dilution series. Initial fluorescence readings of samples with the highest AQP concentrations exhibited significant deviations, leading to their exclusion from subsequent analysis for all AQP2 and AQP5 constructs (both full-length and C-terminal peptides).

### 4.7. MST Data Analysis

Data were exported using the MO.Affinity Analysis software (v. 2.3, NanoTemper Technologies GMBH, Munich, Germany) and curve fitting was done using Origin (v2019, OriginLab, Northampton, MA, USA). The data could be described by a one-to-one binding model:$$y=S1+\left(S2-S1\right)\left(\frac{L_{Free}}{L_{Free}+K_{d}}\right)$$
$$L_{Free}=0.5(L_{Tot}-P_{Tot}-K_{d})+\sqrt{0.25{\left(K_{d}+P_{Tot}-L_{Tot}\right)}^{2}+L_{Tot}K_{d}}$$
where *S*1 and *S*2 are the signals of the unbound and bound form, respectively. *L_Free_* is the free monomeric [AQP] and *L_Tot_* the total monomeric [AQP]. *P_Tot_* is the total [FERM-Alexa Fluor 488] and *K_d_* the dissociation constant. The statistical significance of differences between *K_d_*-values was evaluated using a Z-test for population means. Differences were considered significant if *p* < 0.05.
$$Z=\frac{\left(\bar{X_{1}}-\bar{X_{2}}\right)}{\sqrt{\sigma_{1}^{2}+\sigma_{2}^{2}}}$$

### 4.8. In Silico Modelling of FERM-AQP Complexes

Modelling of molecular interactions between full-length AQP2 and AQP5 or their C-terminal domains (from residues F224 and F226 respectively) and the FERM-domain of human Ezrin was done using ColabFold [41]. Top-scoring predictions were compared and correlated with previously shown interactions between FERM-domains and other peptides (PDB IDs: 1EF1, 1J19, 2D10, 2D11, 1SGH, 2YVC, 2EMT, 2ZPY).

## 5. Conclusions

Our studies show that AQP2 and AQP5 bind the ezrin FERM-domain with similar affinity and suggest that this interaction involves a common novel binding mode in which two regions within the AQP C-termini interact with distinct sites on FERM. Further studies will be needed to validate the interaction and elucidate if this is a unique feature or a common mode of binding for the direct interaction between FERM and membrane proteins during trafficking. Together with the increasing number of ERM-binding partners that are being identified, this will significantly increase our understanding of how competitive and/or cooperative binding of multiple partners is involved in ERM-coordinated membrane-related processes.

## Figures and Tables

**Figure 1 ijms-25-07672-f001:**
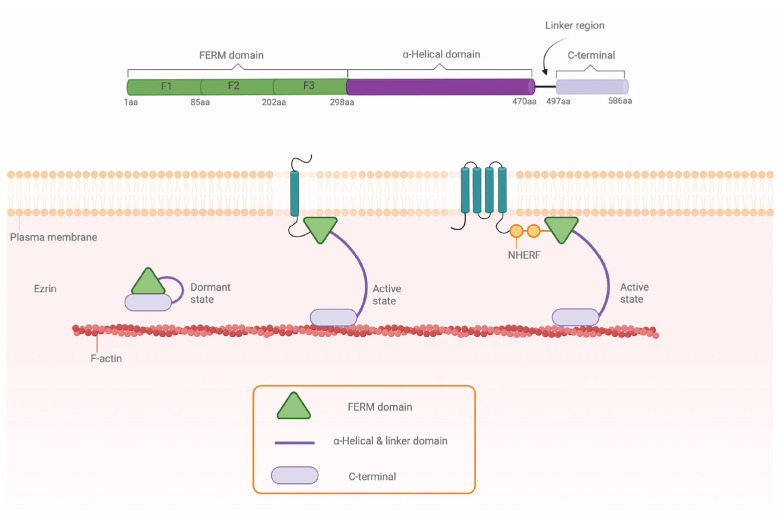
Domain organization and structure of ERM-proteins. ERM-proteins are composed of an N-terminal FERM-domain that can be further divided into F1-F3 subdomains (green), an α-linker domain (purple) and a C-terminal domain (light purple). The residue numbering corresponds to human ezrin. The C-terminal domain binds F-actin while the FERM-domain binds membrane proteins directly or via adaptor proteins such as NHERF. In the dormant state, the C-terminal domain interacts with FERM thereby masking the actin and membrane protein binding sites (adapted from [1]).

**Figure 2 ijms-25-07672-f002:**
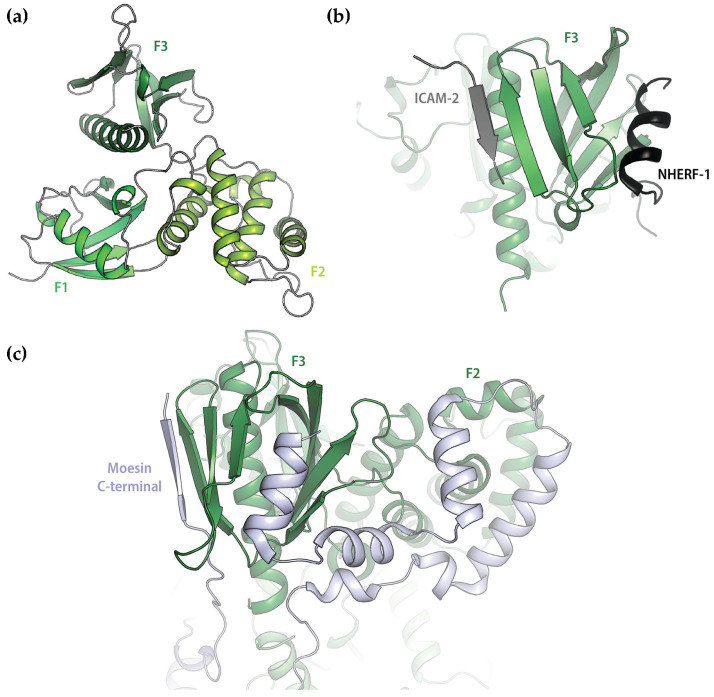
Domain organization and structure of ERM-proteins. (**a**) Crystal structure of the human ezrin FERM-domain (PDB:1NI2) showing F1–F3 subdomains and (**b**) in complex with interacting peptides from NHERF (black, PDB:2D10) and ICAM-2 (grey, PDB:1J19). (**c**) Crystal structure of the auto-inhibitory complex between Moesin-FERM (green) and the Moesin C-terminal domain (light blue) (PDB:1EF1) showing binding to both binding sites on the F3 subdomain.

**Figure 3 ijms-25-07672-f003:**
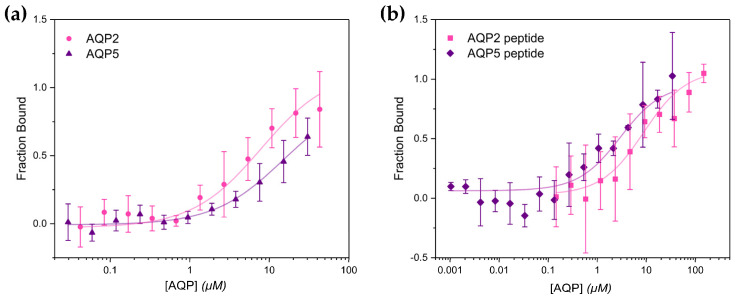
MST analysis of the interaction between AQP2/AQP5 and the Ezrin FERM-domain. (**a**) Binding curves for full-length AQP2 (pink circles) and AQP5 (purple triangles). Curve fitting using a one-to-one binding model resulted in a K_d_ of 7.8 ± 3.8 μM (R^2^ = 0.94%) for AQP2 and 14 ± 5.7 μM for AQP5 (R^2^ = 0.95%). (**b**) Binding curves for AQP2 (pink squares) and AQP5 (purple rhombuses) C-terminal peptides. Curve fitting using a one-to-one binding model resulted in a K_d_ of 8.7 ± 2.4 μM (R^2^ = 0.96%) for AQP2 and 2.9 ± 0.89 μM for AQP5 (R^2^ = 0.97).

**Figure 4 ijms-25-07672-f004:**
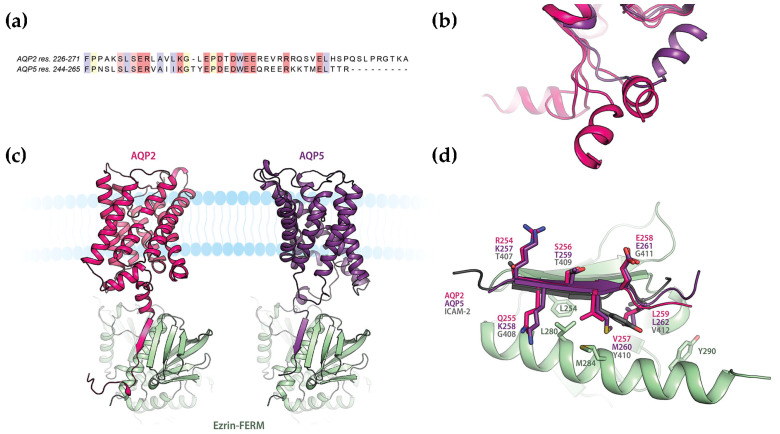
In silico modelling of complexes between full-length AQP2/AQP5 and FERM. (**a**) Sequence alignment of the AQP2 and AQP5 C-termini showing the high sequence conservation. The alignment was made in ClustalW. Residues are color-coded as follows: hydrophobic—blue, polar—light red, charged—dark red, glycine—yellow. (**b**) Structural overlay of crystal structures of human AQP5 (purple, PDB:3D9S) and human AQP2 (pink, PDB:4NEF) showing the conservation of the amphipathic C-terminal helix and the flexibility of its position. In AQP2, the C-terminal helix adopts different positions in all four monomers, none of which is conserved in AQP5. (**c**) Models of the full-length AQP2- and AQP5-FERM complexes generated by ColabFold. In both models, the C-terminus forms a β-strand that complements the β-sheet in the FERM F3 subdomain. (**d**) Zoom-in on the binding site showing how hydrophobic residues on the AQP2/AQP5 β-sheet fit into a hydrophobic groove on F3 in a similar manner as observed in the FERM-ICAM-2 complex (PDB:1J19).

**Figure 5 ijms-25-07672-f005:**
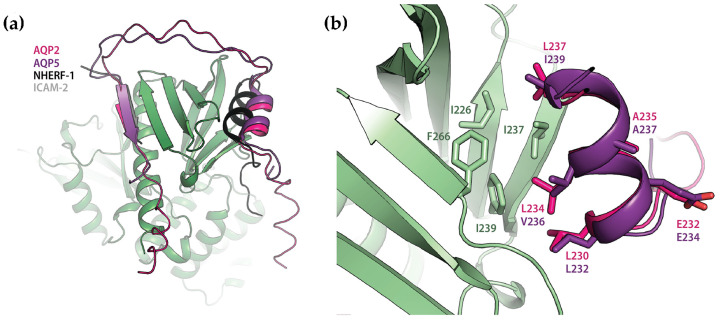
Modelling of complexes between the AQP2/AQP5 C-termini and FERM. (**a**) Structural overlay of the AQP2/AQP5 FERM models showing that the C-terminus forms an α-helix, in addition to the β-strand observed in the full-length complexes. The α-helix binds in a similar manner as seen in crystal structure of the FERM-NHERF-1 complex (black, PDB:2D10) while the β-strand interaction resembles the FERM-ICAM-2 complex (grey, PDB:1J19) as well as the full-length models (Figure 4). (**b**) Zoom in on the α-helix showing amphipathic character and how the hydrophobic side binds into a hydrophobic groove on FERM.

**Figure 6 ijms-25-07672-f006:**
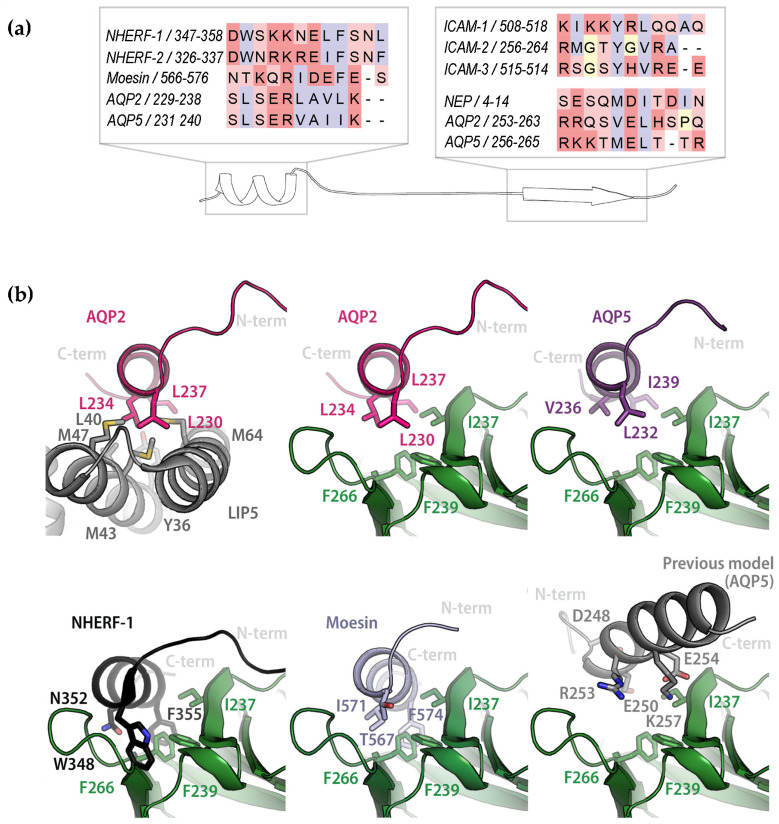
Structural features of FERM-interacting sequences. (**a**) Sequence alignment of proteins binding to FERM. All sequences are of human origin. The secondary structure of the binding regions is either an α-helix (left) or a β-strand (right). In both cases, there are clear similarities between the sequences. The alignment was made in ClustalW and residues are color-coded as follows: hydrophobic—blue, polar—light red, charged—dark red, glycine and proline—yellow. (**b**) Comparison of how the amphipathic AQP C-terminal helix is proposed to bind interacting proteins. In the docking model of AQP2 and the N-terminal domain of LIP5 as well as the AQP2-FERM and AQP5-FERM complexes presented here, the helix binds with its hydrophobic residues into a hydrophobic groove in a highly similar fashion. For the AQP-FERM complexes this closely resembles how NHERF-1 (black, PDB:2D10) and the moesin C-terminus (light blue, PDB:1EF1) interacts with the radixin and moesin FERM-domains, respectively. The analogous interaction is not seen in the previous docking model between AQP5 and FERM (grey).

## Data Availability

The original contributions presented in the study are included in the article/Appendix A, further inquiries can be directed to the corresponding author.

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
