# Peer review of "Structural Basis for the Interaction between the Ezrin FERM-Domain and Human Aquaporins"

_ijms, 2024, doi:10.3390/ijms25147672_

Round 1

Reviewer 1 Report (Previous Reviewer 1)

Comments and Suggestions for Authors

I appreciate that authors have dedecated time and effort revising the manuscript. I am satisfied that the authors have addressed all my concerns in a respnsible manner and I recommend accepting the manuscript in its current form.

Author Response

I appreciate that authors have dedecated time and effort revising the manuscript. I am satisfied that the authors have addressed all my concerns in a respnsible manner and I recommend accepting the manuscript in its current form.

Response: We thank the reviewer for his positive comments and recommendation

Reviewer 2 Report (New Reviewer)

Comments and Suggestions for Authors

The version of the manuscript that I have received is manifestly an improved version, probably by the authors and/or the publisher. So I only have two comments to add.

One, the absence of a brief paragraph with the CONCLUSIONS (as point 4, behind the Discussion) the experimental part would remain as point 5.

Two, check the insertion of a blank space between numerical data and units, except for %) in particular in the experimental part, but also on line 144. On line 502, the number 2 of the calcium chloride formula must be written as a subscript.

Author Response

The version of the manuscript that I have received is manifestly an improved version, probably by the authors and/or the publisher.

Response: We thank the reviewer for the positive comments about our manuscript.

So I only have two comments to add.

One, the absence of a brief paragraph with the CONCLUSIONS (as point 4, behind the Discussion) the experimental part would remain as point 5.

Response: As suggested, we now include a separate conclusion paragraph as point 4. 

Two, check the insertion of a blank space between numerical data and units, except for %) in particular in the experimental part, but also on line 144. On line 502, the number 2 of the calcium chloride formula must be written as a subscript.

Response: Thank you for pointing this out, we have corrected these mistakes.

This manuscript is a resubmission of an earlier submission. The following is a list of the peer review reports and author responses from that submission.

Round 1

Reviewer 1 Report

Comments and Suggestions for Authors

In this manuscript, Strandberg et al. investigated the interaction between aquaporins and the Ezrin FERM domain. The authors employed microscale thermophoresis and computational structural methods to probe the interaction. The authors presents convincing data and proposed a model for FERM/AQP interactions consistent with previous publications. However, I have two concerns about the data presented in this paper and the authors will need to address these before I could consider this manuscript suitable for publication on IJMS:

(1)     The MST analysis on AQP2/5 C-terminal peptides to FERM domain is not reliable. As shown on fig. 3b, both MST fits are very noisy and R2 values are 70.48% and 81.07% for AQP2 and AQP5, respectively, indicating unreliable fit. As a result, KD values reported here is not reliable and may be misleading, and it may be the reason why full-length and peptide MST shows different trends. The authors should repeat the experiment and obtain a less noisy result for a more reliable fit.

(2)     The authors did structural predictions with alphafold and found that the interaction between full-length AQP in complex with FERM did not involve the previously reported interaction pattern where a C-terminal amphipathic helix is involved. The authors argued that the full-length AQPs caused steric hindrance and then used the C-terminal peptide for complex prediction. This approach introduces another critical issue, as wo whether the results have any physiological relevance, since the AQPs C-terminal region does not exist on its own in vivo. I believe a better way to approach this issue is to tweak the ColabFold (and Alphafold) parameters so that the prediction results are more in line with previous predictions and experimental results.

Reviewer 2 Report

Comments and Suggestions for Authors

This study addresses protein interactions between the ERM family which is related to membrane remodeling and the human aquaporin (AQP) proteins that are water channels embedded in the cell membrane. It is mainly constituted by two parts: microscale thermophoresis (MST) study on recombinant proteins to determine their binding affinities; and in-silico protein complexes modelling by a software named ColabFold. The authors found that previously unpublished binding mode between the AQP2 / AQP5 protein and the FERM protein, which belongs to the ERM family. The binding mode involves 2 binding sites in FERM in combination, and previously they were only found individually.

This manuscript would be of high interest to the field of cell biology, structure biology related to cell membrane regulation. The authors provided detailed methods for readers to understand and repeat their experiments. I would recommend its acceptance for publication.

Comments on the Quality of English Language

Please double check if there are any monir language problems, for example

line 141: add "been" at the end;

line 143: remove "for"